# Association of dietary vitamin E intake with peripheral arterial disease: A retrospective cross-sectional study

Qiang Liu [1,2], Xing Wu[3,4], Yun Wang[1,2], Xiang Wang[1,2], Fei Zhao[1,2], Jianjun Shi[1,2]*

1 Department of Cardiovascular Surgery, Taiyuan Central Hospital, Taiyuan, Shanxi, China, 2 Department of Cardiovascular Surgery, The Ninth Clinical College Affiliated to Shanxi Medical University, Taiyuan, Shanxi, China, 3 Department of Nephrology, Taiyuan Central Hospital, Taiyuan, Shanxi, China, 4 Department of Nephrology, The Ninth Clinical College Affiliated to Shanxi Medical University, Taiyuan, Shanxi, China

* 13934282361@163.com

## Abstract

### Background

The relationship between the amount of dietary vitamin E consumed and the development of peripheral arterial disease (PAD) remains a topic of debate. This relationship is of the utmost importance in the realms of healthcare and public health and is currently a highly researched and prominent topic. This study aimed to present the relationship between dietary vitamin E intake and PAD.

### Methods

In a retrospective cross-sectional analysis, data from 6,588 participants in the National Health and Nutrition Examination Survey of the United States were examined during the period 1999–2004. We collected data on age, sex, race, marital status, education, physical activity, income, smoking, hypertension, diabetes, cardiovascular disease, body mass index, total cholesterol and HbA1c. Logistic regression and smooth curve fitting were used to support the research objectives.

### Results

After accounting for all relevant factors, a negative correlation between dietary vitamin E intake and the likelihood of PAD was observed (OR: 0.981, 95% CI: 0.957–1.004). The overall prevalence of PAD was 5.9%, with 49.6% in males and 50.4% in females. Individuals in the third quartile of dietary vitamin E intake had a lower occurrence rate of peripheral artery disease than those in the first quartile (OR: 0.68, 95% CI: 0.51, 0.91). Similar patterns of association were observed in the subgroup analysis (all P values for interaction were > 0.05).

**Data availability statement:** All NHANES data and information are publicly available at https://www.cdc.gov/nchs/nhanes/.

**Funding:** The author(s) received no specific funding for this work.

## Conclusions

Our study suggests a negative association between dietary vitamin E intake and incidence of PAD. Therefore individuals with insufficient dietary vitamin E intake, especially those with a very low intake, should consider increasing their vitamin E intake to lower the risk of developing PAD. These findings should be considered when offering dietary guidance and nutrition education to prevent PAD.

## Introduction

Peripheral arterial disease (PAD) is caused by atherosclerosis, a condition characterized by the accumulation of fatty deposits in the arteries of the lower extremities and feet [1]. It is prevalent disorder affecting over 200 million individuals worldwide [2]. In the United States, PAD affects around 8.5 million adults aged 40 years and above, with similar occurrence rates among men and women [3,4]. The economic and health burdens of PAD are substantial, emphasizing the need to identify its causes and implement appropriate measures. Previous studies have highlighted the significant roles of inflammatory markers, endothelial dysfunction, and oxidative stress in PAD development [5,6]. Although smoking, diabetes, hypertension, and dyslipidemia are established risk factors for PAD [3,7], the influence of diet is unclear. Further investigations are warranted to enhance our understanding of preventive measures and management strategies for PAD, given the limited knowledge of effective preventive factors.

Vitamin E, an antioxidant that dissolves in fat, is recognized as a key player in reducing lipid peroxidation in experimental settings [8]. Gey [4] first proposed the idea of vitamin E's potential in preventing cardiovascular disease in the "antioxidant hypothesis of atherosclerosis". Subsequent evidence linked vitamin E to atherosclerosis and thrombotic complications, attributing its anti- atherosclerotic activity to a range of biological functions, including antioxidant functions as scavengers of free radicals, in addition to non-antioxidant roles, such as controlling signal transduction, cellular proliferation, and gene expression. Vitamin E has attracted interest because of its anti-atherogenic and antioxidant properties, which have been associated with improved cardiovascular health in numerous studies. There was a significant increase in alpha-tocopherol levels [9–11]. However while some studies indicate that supplemental vitamin E may offer protection against cardiovascular diseases, others do not, and a few even suggest the potential for adverse effects [12,13].

Therefore, to address this knowledge gap, the ultimate goal was to present a deeper understanding of how dietary vitamin E intake influences PAD and the implications of this relationship. We sought to gain deeper insights into the relationship between dietary vitamin E intake and PAD through a retrospective cross-sectional study involving 6,588 participants in the National Health and Nutrition Examination Survey (NHANES). We aimed to provide valuable information for personalized nutritional management strategies and contribute to a deeper understanding of the nutritional status of individuals and populations.

## Methods

### Study population

During a retrospective cross-sectional analysis, data from 6,588 participants in the NHANES of the United States were examined during the period 1999–2004 [14–16]. The NHANES compiles diverse health-related data, including demographic details, physical examination outcomes, laboratory results, and dietary patterns. The National Center for Health Statistics

collected this information with the approval of its ethics review board. Before participating in the NHANES, all participants provided written informed consent [17,18]. NHANES data can be accessed by the public through the NHANES website (http://www.cdc.gov/nchs/nhanes.htm). In this study, researchers independent of the original NHANES program conducted an analysis of publicly available data sourced from NHANES records. As this involves a secondary analysis of publicly accessible data, ethics approval is not required. Of the 31,126 individuals who participated in the in-home interview, we excluded those aged below 40 years (n = 21,156) because an ankle brachial index (ABI) was not measured in this age group. Participants were excluded from the study if they had missing bilateral ABI data (n = 2408), ABI values exceeding 1.4 (n = 65) or were lacking diet data (n = 169). Individuals with missing covariates were excluded (n = 740), resulting in a final analysis sample of 6,588 participants.

## Ankle-brachial index

Systolic blood pressure measurements were taken from the right arm (brachial artery) and both ankles (posterior tibial artery) after a brief period of rest. If the participant's right arm measurements were unavailable due to any conditions that could affect accuracy, such as open wounds or dialysis shunts, the left arm was used for the brachial pressure measurement. Individuals aged 40-59 had their systolic blood pressure measured twice, while those aged 60 and older had it measured only once. The ankle-brachial index (ABI) was calculated by dividing the average systolic blood pressure (ASBP) of the ankle by the ASBP in the arm. For participants aged 60 and older, the single reading represented the mean values. ABI < 0.9 was used to define PAD [19,20].

## Dietary vitamin E intake

Data related to vitamin E intake in diets was gathered from the NHANES 1999 to 2004 first-day dietary interview examination files. A 24-hour dietary recall interview was conducted via the NHANES computer-assisted dietary data interview system to acquire detailed dietary intake data for all participants. This interview captured details of all foods and beverages consumed within a 24-hour duration, such as consumption time, eating occasion, food descriptions, portion sizes, food sources, and location of consumption. Following the dietary recall, a set of health-related inquiries were made. The data collection approach utilized the Automated Multiple Pass Method, which is the dietary data collection tool of the US Department of Agriculture (USDA). Detailed methodologies for the dietary survey can be found in the NHANES Dietary Interviewer's Procedure Manual [15]. While 24-h dietary recalls possess inherent limitations in terms of reliability and effectiveness, they offer a greater level of specificity regarding the varieties and amounts of ingested foods than food frequency questionnaires [21–23]. The participants were then divided into tertile (Q1–Q3) based on their dietary vitamin E consumption.

## Covariates

Standardized questionnaires were administered to obtain socio-demographic and lifestyle information. The Mobile Examination Center furnished the results of the examinations, including body mass index (BMI), blood pressure, and various biochemical parameters. Different potential covariates were evaluated using the available literature [24], including age, sex, race/ethnicity, education level, marital status, family income, smoking status, physical activity, BMI, laboratory parameters (total cholesterol and glycated hemoglobin A1c), and comorbidities (e.g., diabetes, hypertension, and cardiovascular disease [CVD]). For detailed information regarding these variables, such as the methods used for measurement, questionnaire data, and

the list of variables, please refer to the official NHANES website (www.cdc.gov/nchs/nhanes/ accessed on 1 May 2022). The categorization of family income was based on the poverty income ratio (PIR) into three groups: low (PIR ≤ 1.3), medium (PIR > 1.3–3.5), and high (PIR > 3.5). Smoking status was classified according to established definitions from previous studies: never smoked, current smoker, and former smoker. To evaluate physical activity levels, participants were asked about intense exercises leading to notable changes in breathing and heart rate (such as swimming or high-speed cycling), as well as moderate exercises causing mild-to-moderate increases in these physiological parameters (for example, golfing or leisurely biking). The duration of each activity should have been at least 10 min within the last month. Physical activity levels were categorized into three groups: below-moderate (lacking both moderate and intense activities), moderate (absence of intense exercise but inclusion of at least one moderately active pursuit), and high-intensity (presence of at least one episode involving rigorous exertion).

The identification of previous illnesses (such as hypertension and diabetes) was established through a questionnaire by asking participants if they had previously informed a doctor about their medical conditions. The history of cardiovascular disease was determined based on individuals' self-reported experiences with congestive heart failure, coronary heart disease, angina pectoris, heart attack, and stroke. BMI was calculated using a conventional method that considers weight and height.

## Statistical analysis

This study used a rigorous and comprehensive approach to assess the effects of dietary vitamin E intake on PAD. Analysis of the association between dietary vitamin E intake and PAD was conducted using robust statistical techniques, such as binary logistic regression models that provide odds ratios and 95% confidence intervals, consideration of confounders based on expert judgment, existing scientific literature, and identification of significant covariates in univariate analysis, indicating a thoughtful approach to controlling potential sources of bias. Additionally, the use of a subgroup analysis to examine the relationship between vitamin E intake and PAD within specific subgroups adds to these findings. The handling of missing data through listwise deletion, sensitivity analyses to assess the robustness of the findings, and the reporting and comparison of effect sizes and p-values from various association inference models further enhanced the credibility and reliability of the study's conclusions. Overall, the methodology and analytical approach contributed to the robustness and validity of the findings.

Statistical analyses were performed using R Statistical Software (Version 4.2.2, http://www.R-project.org, The R Foundation) and the Free Statistics Analysis Platform (Version 1.9, Beijing, China, http://www.clinicalscientists.cn/freestatistics). A detailed explanation of their functionalities ensured transparency and reproducibility of the analytical methods used in this study. Overall, the statistical analysis appears to be comprehensive, well executed, and supported by appropriate software tools, contributing to the strength of the study's findings.

## Results

### Characteristics of the study population

In total, 31,126 individuals participated in the in-home interviews. Among them, 21,156 participants aged < 40 years were excluded. Furthermore, participants who had missing bilateral ABI data (n = 2408), ABI values exceeding 1.4 (n = 65), or lacking diet information (n = 169) were excluded. Participants with incomplete covariate data were excluded (n = 740), resulting in a final sample size of 6,588 participants (Fig 1).

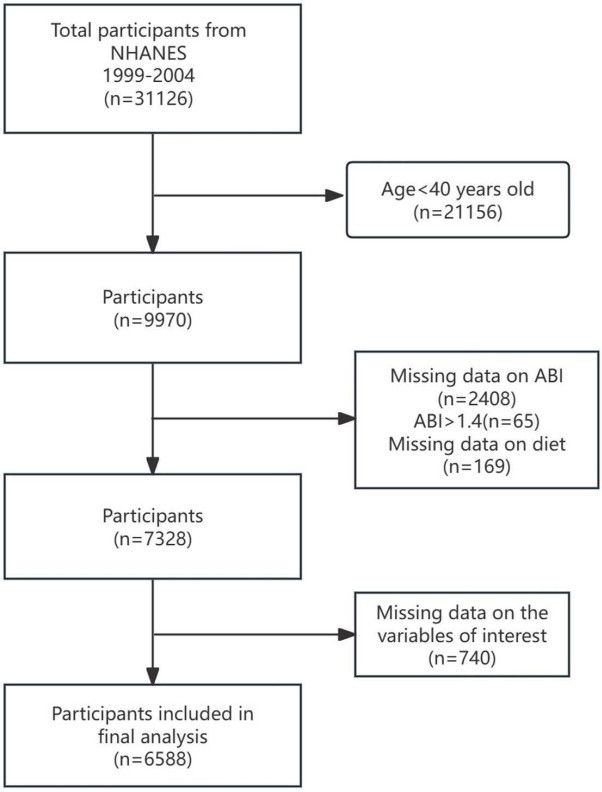

**Fig 1. Research flowchart.**

The study included qualified individuals aged 40 years or older with a PAD prevalence rate of 5.9%. Table 1 presents the basic features of the participants categorized according to their dietary consumption of vitamin E. The three groups differed in terms of sex, age, education level, race, MS, PIR, physical activity, HbA1c, smoking status, hypertension, CVD, and diabetes mellitus (P < 0.05). However, the distribution of patient characteristics, such as BMI and total cholesterol, among the dietary vitamin E intake groups was similar (P > 0.05) (Table 1).

## Association between dietary vitamin E intake and PAD

In the multivariable logistic regression analyses, it was found that participants in the Q3 group exhibited a lower probability of PAD (OR = 0.68, 95% CI = 0.51–0.91, P = 0.009) than the Q1 group, after adjusting for potential confounders (Table 2, model 5).

The adjusted smoothed plots suggested a straightforward linear relationship between dietary vitamin E intake and PAD (Fig 2; P for nonlinearity = 0.897; the highest and lowest 0.5% were trimmed for each dietary vitamin E intake measure). As the dietary vitamin E intake increased, the risk of PAD decreased.

Fig 2 shows the correlation between dietary vitamin E intake and PAD in both males and females. Data were analyzed using a multivariate logistic regression model with restricted cubic splines. To account for various factors, such as sex, age, BMI, severity of illness, and reason for admission (Model 5), dietary vitamin E intake was considered a continuous variable. The reference standard was the median dietary vitamin E intake.

**Table 1. Baseline characteristics stratified by the dietary vitamin E intake quartiles (Q).**

| Variables | Total (n = 6588) | Q1 (n = 2191) | Q2 (n = 2195) | Q3 (n = 2202) | p-Value |
|---|---|---|---|---|---|
| **Gender, n (%)** | | | | | < 0.001 |
| Male | 3362 (51.0) | 951 (43.4) | 1086 (49.5) | 1325 (60.2) | |
| Female | 3226 (49.0) | 1240 (56.6) | 1109 (50.5) | 877 (39.8) | |
| **Age (years)** | 60.1 ± 13.0 | 61.6 ± 12.9 | 60.2 ± 13.2 | 58.4 ± 12.6 | < 0.001 |
| **Race, n (%)** | | | | | < 0.001 |
| Mexican American | 1371 (20.8) | 507 (23.1) | 449 (20.5) | 415 (18.8) | |
| Other Hispanic | 261 (4.0) | 93 (4.2) | 84 (3.8) | 84 (3.8) | |
| Non-Hispanic white | 3635 (55.2) | 1070 (48.8) | 1238 (56.4) | 1327 (60.3) | |
| Non-Hispanic black | 1127 (17.1) | 453 (20.7) | 355 (16.2) | 319 (14.5) | |
| Other races | 194 (2.9) | 68 (3.1) | 69 (3.1) | 57 (2.6) | |
| **Education level, n (%)** | | | | | < 0.001 |
| Less than high school | 1140 (17.3) | 513 (23.4) | 352 (16) | 275 (12.5) | |
| High school diploma or GED | 2562 (38.9) | 908 (41.4) | 854 (38.9) | 800 (36.3) | |
| More than high school | 2886 (43.8) | 770 (35.1) | 989 (45.1) | 1127 (51.2) | |
| **MS, n (%)** | | | | | < 0.001 |
| Married/Living with partner | 4381 (66.5) | 1316 (60.1) | 1476 (67.2) | 1589 (72.2) | |
| Widowed/Divorced/Separated | 1839 (27.9) | 746 (34) | 604 (27.5) | 489 (22.2) | |
| Never married | 368 (5.6) | 129 (5.9) | 115 (5.2) | 124 (5.6) | |
| **PIR, n (%)** | | | | | < 0.001 |
| Low income | 1504 (22.8) | 635 (29) | 490 (22.3) | 379 (17.2) | |
| Medium income | 2290 (34.8) | 785 (35.8) | 764 (34.8) | 741 (33.7) | |
| High income | 2794 (42.4) | 771 (35.2) | 941 (42.9) | 1082 (49.1) | |
| **Physical activity, n (%)** | | | | | < 0.001 |
| Sedentary | 3017 (45.8) | 1190 (54.3) | 989 (45.1) | 838 (38.1) | |
| Moderate | 2071 (31.4) | 629 (28.7) | 706 (32.2) | 736 (33.4) | |
| Vigorous | 1500 (22.8) | 372 (17) | 500 (22.8) | 628 (28.5) | |
| **BMI, kg/m2** | 28.5 ± 5.6 | 28.3 ± 5.5 | 28.4 ± 5.6 | 28.7 ± 5.6 | 0.135 |
| **Total cholesterol, mg/dL** | 209.6 ± 41.3 | 210.9 ± 41.9 | 209.0 ± 41.0 | 208.7 ± 41.0 | 0.150 |
| **HbA1c, %** | 5.8 ± 1.1 | 5.8 ± 1.2 | 5.8 ± 1.0 | 5.7 ± 1.1 | 0.003 |
| **Diabetes, n (%)** | | | | | |
| No | 5742 (87.2) | 1864 (85.1) | 1921 (87.5) | 1957 (88.9) | |
| Yes | 846 (12,8) | 327 (14.9) | 274 (12,5) | 245 (11.1) | |
| **Hypertension, n (%)** | | | | | 0.012 |
| No | 4251 (64.5) | 1361 (62.1) | 1432 (65.2) | 1458 (66.2) | |
| Yes | 2337 (35.5) | 830 (37.9) | 763 (34.8) | 744 (33.8) | |
| **CVD, n (%)** | | | | | 0.025 |
| No | 5546 (84.2) | 1809 (82.6) | 1854 (84.5) | 1883 (85.5) | |
| Yes | 1042 (15.8) | 382 (17.4) | 341 (15.5) | 319 (14.5) | |
| **Smoking, n (%)** | | | | | < 0.001 |
| Never | 3043 (46.2) | 1011 (46.1) | 1038 (47.3) | 994 (45.1) | |
| Former | 2288 (34.7) | 707 (32.3) | 753 (34.3) | 828 (37.6) | |
| Current | 1257 (19.1) | 473 (21.6) | 404 (18.4) | 380 (17.3) | |
| **PAD, n (%)** | | | | | < 0.001 |
| No | 6199 (94.1) | 2023 (92.3) | 2054 (93.6) | 2122 (96.4) | |
| Yes | 389 (5.9) | 168 (7.7) | 141 (6.4) | 80 (3.6) | |

Values for BMI and age Values are given as mean ± standard deviation, other values are presented as numbers and percentages. BMI, body mass index; CVD, cardiovascular disease; CRP, C-reactive protein; HbA1c, glycosylated hemoglobin; MS, marital status; PIR, poverty income ratio; PAD, peripheral arterial disease.

**Table 2. Odds ratios (ORs) and 95% confidence interval (CI) of the dietary vitamin E intake quartiles for PAD.**

| Variable | Vitamin E(mg) | P-value | Q1 (0-4.4) | Q2 (4.41-7.69) | P-value | Q3 (7.7-97.0) | P-value | Trend. test | P-value |
|---|---|---|---|---|---|---|---|---|---|
| | OR (95%CI) | | OR (95%CI) | OR (95%CI) | | OR (95%CI) | | | |
| Model 1 | 0.944(0.92~0.969) | <0.001 | 1 (Ref) | 0.83 (0.66~1.04) | 0.108 | 0.45 (0.35~0.6) | <0.001 | 0.69 (0.61~0.79) | <0.001 |
| Model 2 | 0.973 (0.949~0.997) | 0.0279 | 1 (Ref) | 0.97 (0.76~1.23) | 0.779 | 0.63 (0.47~0.84) | 0.002 | 0.81 (0.71~0.93) | 0.003 |
| Model 3 | 0.975 (0.951~0.999) | 0.0406 | 1 (Ref) | 0.97 (0.76~1.24) | 0.808 | 0.64 (0.48~0.86) | 0.003 | 0.82 (0.71~0.94) | 0.005 |
| Model 4 | 0.976 (0.952~1) | 0.0517 | 1 (Ref) | 0.97 (0.76~1.24) | 0.804 | 0.65 (0.48~0.86) | 0.003 | 0.82 (0.71~0.94) | 0.005 |
| Model 5 | 0.981 (0.957~1.004) | 0.1075 | 1 (Ref) | 1 (0.78~1.28) | 0.992 | 0.68 (0.51~0.91) | 0.009 | 0.84 (0.73~0.97) | 0.016 |

Data are presented as odds ratios, 95% CIs (confidence intervals), and p-value.

Model 1 adjusted for none.

Model 2 adjusted for age, gender, race, education level, MS, physical activity.

Model 3 adjusted for age, gender, race, education level, MS, physical activity, PIR, BMI, total cholesterol, HbA1c.

Model 4 adjusted for age, gender, race, education level, MS, physical activity, PIR, BMI, total cholesterol,HbA1c, diabetes, hypertension, CVD.

Model 5 adjusted for all covariates.

Shaded areas represent 95% confidence intervals. The curves highlight the inverse relationship between dietary vitamin E intake and PAD when considering low or near-median values of dietary vitamin E intake. Owing to limited patient numbers and the nature of the cubic fit analysis, there was significant variability in the 95% confidence intervals at extreme values.

## Stratification analysis

In this study, we performed a subgroup analysis based on various factors, including age, sex, ethnicity, presence of hypertension or diabetes, educational background, history of CVD, income level (PIR), physical activity level, and smoking status. Additionally, sensitivity and supplementary analyses were conducted, and the results of the subgroup analysis indicated no significant evidence of effect modification or interaction based on common risk factors for PAD (all P values for interaction were >0.05) (Fig 3).

## Discussion

To the extent of our understanding, in a retrospective cross-sectional study conducted on a significant number of participants involved in the NHANES between 1999 and 2004 in the United States, we consistently observed an inverse correlation between the intake of dietary vitamin E and likelihood of developing PAD. Results were consistent across clinical subgroups and in sensitivity analyses. These observations would have important implications for current PAD management strategies.

Vitamin E, a membrane-bound antioxidant found in nuts, vegetable oils, and fish, is associated with reduced cellular damage in cardiovascular disease [21]. Its antioxidant and anti-inflammatory properties, along with its ability to prevent platelet aggregation and smooth cell proliferation, have been shown to exhibit effects that counteract the development of atherosclerosis [25]. Our research findings align with previous observational studies indicating an inverse relationship between the dietary intake of vitamin E and the occurrence of PAD among participants in the NHANES conducted from 1999 to 2004 [9,14,16]. According to a study conducted by Lane et al. [21], an increased intake of vitamin E exhibited a notable safeguarding effect, regardless of the presence of conventional cardiovascular risk factors, within the population of the United States. Not come singly but in pairs, Antonelli-Incalzi et al. [26] conducted a study on the nutrient consumption of

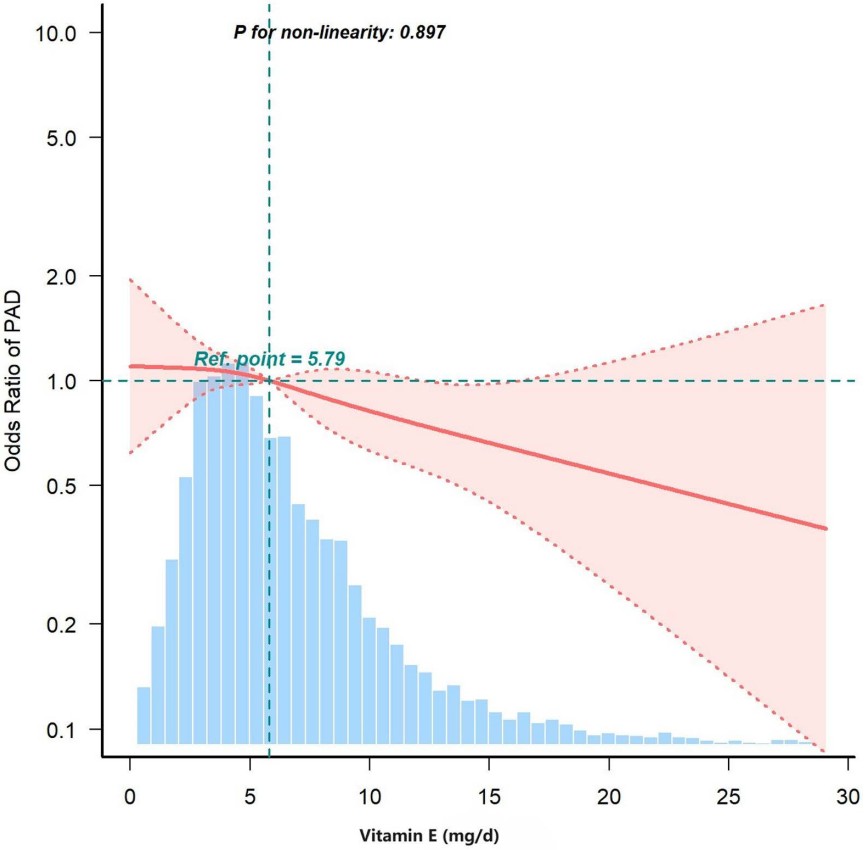

**Fig 2. Association between dietary vitamin E intake and PAD odds ratio.** Solid and dashed lines represent the predicted value and 95% confidence intervals. They were adjusted for age, sex, race/ethnicity, education level, marital status, family income, smoking status, physical activity, BMI, total cholesterol, glycated hemoglobin A1c, and comorbidities. Only 99% of the date is shown.

1251 individuals living in their own homes, who were part of the InCHIANTI research. The average age of participants were 68 years. They also investigated whether there was a potential decrease in the risk of PAD associated with a daily intake of Vitamin E equal to or greater than 7.726 mg/d. (OR: 0.37; 95% CI 0.16–0.84). In addition, the Rotterdam Study found that there was a negative correlation between vitamin E consumption and PAD in men. However, it remains unclear whether the association between antioxidant intake and PAD differs between sexes or whether these findings are influenced by the different dietary patterns observed in men than in women [27]. Further research is necessary to validate our results and investigate the intricate relationships and potential mechanisms involved. Our findings support and expand upon previous research conducted on participants in the NHANES from 1999 to 2004 in the United States. We found that a higher intake of dietary vitamin E was associated with a reduced risk of PAD after accounting for all relevant factors (OR, 0.981; 95% CI, 0.957–1.004). When we divided dietary vitamin E intake into quartiles, individuals in the third quartile had a lower incidence rate of PAD than those in the first quartile (OR, 0.68; 95% CI: 0.51, 0.91). Subgroup analysis yielded similar results without any significant differences (all P values for interactions were >0.05). Extensive clinical research has demonstrated the potential protective effects of vitamin E against long-term cardiovascular diseases [28]. Studies have indicated that taking vitamin E supplements for more than four years can result in a substantial 59% decrease in mortality from coronary disease [29]. The discussion

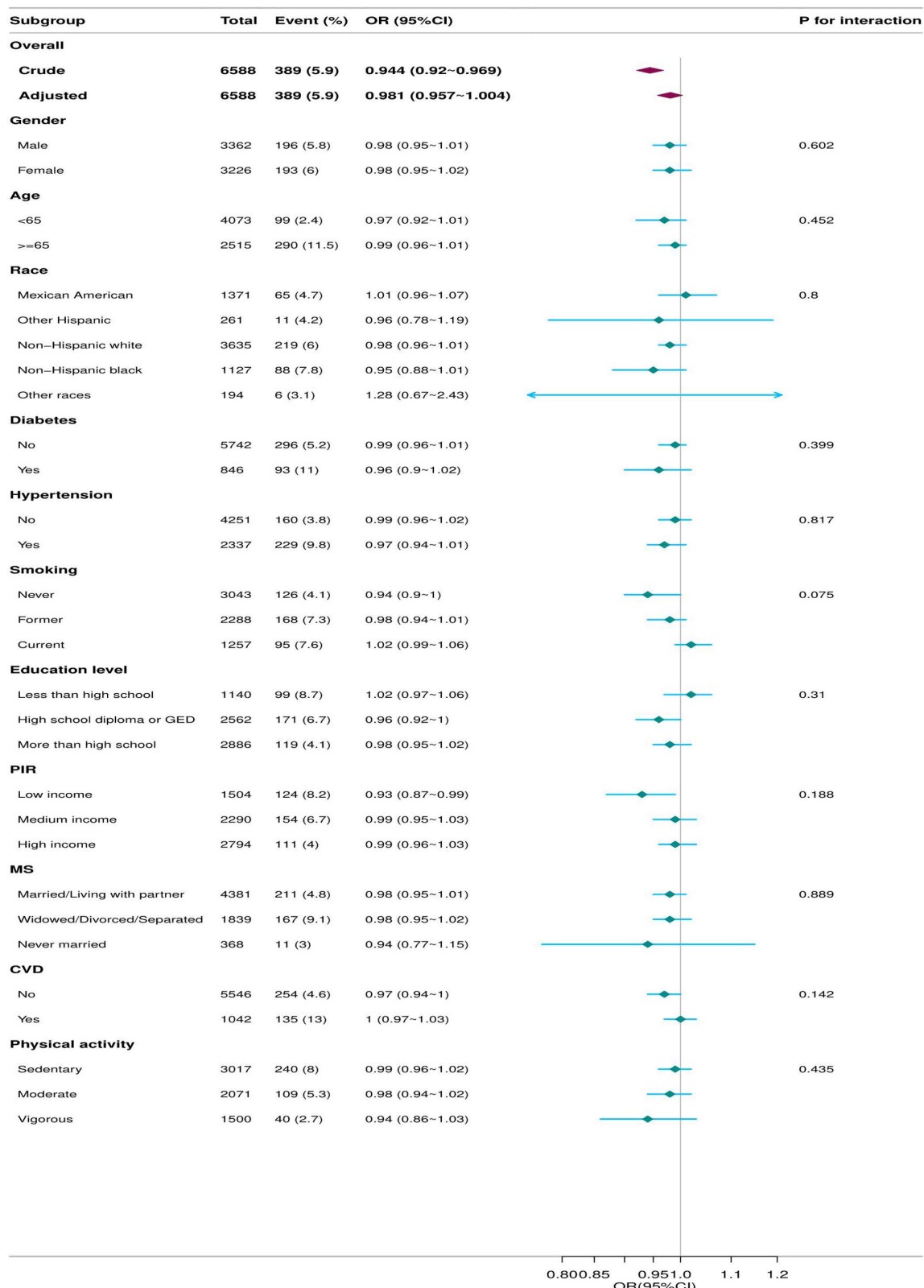

**Fig 3. The relationship between dietary vitamin E intake and PAD according to basic features.** Except for the stratification component itself, each stratification factor was adjusted for all other variables.

regarding the impact of vitamin E on cardiovascular diseases is complex. While certain studies indicate that vitamin E supplements may provide protection against cardiovascular diseases, others do not support this claim. Additionally, a minority of studies have even suggested potential adverse effects [30,31]. So further investigations are necessary to validate our findings and explore the intricate relationships and potential mechanisms involved.

This was a well-designed retrospective cross-sectional study with a robust sample size and stringent inclusion and exclusion criteria. Baseline data on a wide range of lifestyle factors, including dietary cofactors, were collected, enabling adjustment for potential confounders in the analyses. Additionally, a comprehensive set of sensitivity analyses was conducted to enhance the interpretation of the results and validate the main findings.

However, this study had several limitations. First, owing to the cross-sectional and observational nature of the analysis, causal relationships among dietary vitamin E intake, covariates, and PAD could not be definitively established. Despite efforts to control for relevant confounders in the multivariate model, the presence of unmeasured or unknown residual confounders, such as dietary habits and family income, may have led to an overestimation of the observed associations. Furthermore, in this study, the intake of vitamin E was not energy-adjusted. Nonetheless, the study mainly included American adults, which limits the generalizability of the findings and necessitates further research on diverse populations.

## Conclusions

Our study provides supporting evidence indicating a detrimental correlation between the consumption of dietary vitamin E and the occurrence of PAD in US adults older than 40 years. Therefore, individuals with insufficient dietary vitamin E intake, especially those with a very low vitamin E intake, should consider increasing their vitamin E intake to lower the risk of developing PAD. These findings should be considered when offering dietary guidance and nutrition education to prevent PAD.

## Acknowledgments

We express our gratitude to Jie Liu from the Department of Vascular and Endovascular Surgery at Chinese PLA General Hospital for providing valuable assistance in statistical analysis, study design consultations, and offering insightful feedback on the manuscript.

## Author contributions

**Conceptualization:** Jianjun Shi.

**Data curation:** Qiang Liu.

**Funding acquisition:** Yun Wang, Jianjun Shi.

**Methodology:** Qiang Liu.

**Writing – original draft:** Qiang Liu, Xing Wu, Xiang Wang, Fei Zhao.

**Writing – review & editing:** Qiang Liu, Xing Wu.

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
