## [Decision Letter · Decision Letter 0]

1 Nov 2024

PONE-D-24-23667Association of dietary vitamin E intake with peripheral arterial disease: A retrospective cross-sectional studyPLOS ONE

Dear Dr. Liu,

Thank you for submitting your manuscript to PLOS ONE. After careful consideration, we feel that it has merit but does not fully meet PLOS ONE’s publication criteria as it currently stands. Therefore, we invite you to submit a revised version of the manuscript that addresses the points raised during the review process.

We look forward to receiving your revised manuscript.

Kind regards,

Hasan Durmus

Academic Editor

PLOS ONE

Journal Requirements: When submitting your revision, we need you to address these additional requirements. 1. Please ensure that your manuscript meets PLOS ONE's style requirements, including those for file naming. The PLOS ONE style templates can be found at https://journals.plos.org/plosone/s/file?id=wjVg/PLOSOne_formatting_sample_main_body.pdf and https://journals.plos.org/plosone/s/file?id=ba62/PLOSOne_formatting_sample_title_authors_affiliations.pdf 2. We suggest you thoroughly copyedit your manuscript for language usage, spelling, and grammar. If you do not know anyone who can help you do this, you may wish to consider employing a professional scientific editing service.  The American Journal Experts (AJE) (https://www.aje.com/) is one such service that has extensive experience helping authors meet PLOS guidelines and can provide language editing, translation, manuscript formatting, and figure formatting to ensure your manuscript meets our submission guidelines. Please note that having the manuscript copyedited by AJE or any other editing services does not guarantee selection for peer review or acceptance for publication.  Upon resubmission, please provide the following: The name of the colleague or the details of the professional service that edited your manuscript A copy of your manuscript showing your changes by either highlighting them or using track changes (uploaded as a *supporting information* file) A clean copy of the edited manuscript (uploaded as the new *manuscript* file)” 3. Please include your full ethics statement in the ‘Methods’ section of your manuscript file. In your statement, please include the full name of the IRB or ethics committee who approved or waived your study, as well as whether or not you obtained informed written or verbal consent. If consent was waived for your study, please include this information in your statement as well. 4. Please include your tables as part of your main manuscript and remove the individual files. Please note that supplementary tables (should remain/ be uploaded) as separate ""supporting information"" files

Reviewers' comments:

Reviewer's Responses to Questions

**Comments to the Author**

1. Is the manuscript technically sound, and do the data support the conclusions?

Reviewer #1: Yes

Reviewer #2: Partly

2. Has the statistical analysis been performed appropriately and rigorously? 

Reviewer #1: Yes

Reviewer #2: Yes

3. Have the authors made all data underlying the findings in their manuscript fully available?

Reviewer #1: Yes

Reviewer #2: Yes

4. Is the manuscript presented in an intelligible fashion and written in standard English?

Reviewer #1: Yes

Reviewer #2: No

5. Review Comments to the Author

Reviewer #1: This is a population study whose methodology was presented in a coherent and well-founded manner, thereby providing a robust foundation for the authors' findings.

The following considerations pertain to adjustments:

1. The acronym "ABI" should be written in full for the first time it is mentioned in the text.

2. While the methods of the primary study are duly referenced, it is nevertheless necessary to provide a detailed account of the methodology employed for the assessment of vitamin E consumption in this manuscript. Please indicate the number of 24-hour dietary recalls utilized in the study. Please clarify whether vitamin E intake was corrected for energy. It would be beneficial to ascertain whether the participants were taking a vitamin E supplement.

3. As noted, Reference 14 examined the same population and investigated the correlation between vitamin E intake and DAP. In what ways does your study differ from that of Reference 14?

4. The majority of the references utilized in this study are old, with 80% of them published more than five years ago. It is recommended that more current references be cited, particularly those used in the discussion to support the relationship between vitamin E consumption and DAP.

5. In line 228, the text makes reference to "recent studies," yet no corresponding reference is provided. Please provide the references for these recent studies.

6. The observational, cross-sectional study does not define a causal relationship. It is therefore advisable to exercise caution when considering the phrase "These findings should be used when formulating clinical guidelines."

7. The extant literature indicates that the isolated consumption of vitamin E does not provide cardiovascular protection. This is evidenced by the fact that studies that used supplementation did not observe the same results as observational studies. It is therefore of interest that this topic is introduced for discussion.

Reviewer #2: It is a well-written but incomplete article. Language editing is strongly recommended. Professional language support is recommended. In addition to the grammar check, the statistical as well as grammatical reasons why certain methods are favoured should be elaborated.

6. PLOS authors have the option to publish the peer review history of their article (what does this mean? ). If published, this will include your full peer review and any attached files.

**Do you want your identity to be public for this peer review?** For information about this choice, including consent withdrawal, please see our Privacy Policy .

Reviewer #1: No

Reviewer #2: No

---

## [Author Response · Author response to Decision Letter 0]

26 Dec 2024

Dear Editor and Reviewers:

On behalf of my co-authors, we greatly appreciate the careful review and comments from both you and the reviewers. We believe that by implementing the suggested changes, we now have a stronger manuscript entitled “Association of dietary vitamin E intake with peripheral arterial disease: A retrospective cross-sectional study” for submission to Journal of PLOS ONE. We look forward to your positive response to the revised work submitted here.

We present here point-to-point responses for each of the comments in the attached document and have revised our manuscript accordingly. We do not change our statistics or results. And we hope the revised manuscript could be acceptable to you. Revised sections are identified with red text in the paper.

There are no conflicts of interest regarding this work. All authors have read the revised manuscript and approved its submission to the journal “PLOS ONE”. Please do not hesitate to contact us if you have any questions.

Thank you and best regards.

Yours Sincerely,

Jianjun Shi

mail: 13934282361@163.com

Department of Cardiovascular Surgery,Taiyuan Central Hospital; Department of Cardiovascular Surgery, the Ninth Clinical College Affiliated to Shanxi Medical University, Taiyuan,030000, China.

Responds to the reviewer’s comments:

Reviewer #1:

Dear Reviewer,

Thank you very much for your time involved in reviewing the manuscript and your comments have greatly helped to improve the quality of our manuscript.

Comments: This is a population study whose methodology was presented in a coherent and well-founded manner, thereby providing a robust foundation for the authors' findings.The following considerations pertain to adjustments:

1. The acronym "ABI" should be written in full for the first time it is mentioned in the text.

2. While the methods of the primary study are duly referenced, it is nevertheless necessary to provide a detailed account of the methodology employed for the assessment of vitamin E consumption in this manuscript. Please indicate the number of 24-hour dietary recalls utilized in the study. Please clarify whether vitamin E intake was corrected for energy. It would be beneficial to ascertain whether the participants were taking a vitamin E supplement.

3. As noted, Reference 14 examined the same population and investigated the correlation between vitamin E intake and DAP. In what ways does your study differ from that of Reference 14?

4. The majority of the references utilized in this study are old, with 80% of them published more than five years ago. It is recommended that more current references be cited, particularly those used in the discussion to support the relationship between vitamin E consumption and DAP.

5. In line 228, the text makes reference to "recent studies," yet no corresponding reference is provided. Please provide the references for these recent studies.

6. The observational, cross-sectional study does not define a causal relationship. It is therefore advisable to exercise caution when considering the phrase "These findings should be used when formulating clinical guidelines."

7. The extant literature indicates that the isolated consumption of vitamin E does not provide cardiovascular protection. This is evidenced by the fact that studies that used supplementation did not observe the same results as observational studies. It is therefore of interest that this topic is introduced for discussion.

Comment 1:

The acronym "ABI" should be written in full for the first time it is mentioned in the text.

Respond 1:

Thank you for the valuable suggestion. Firstly, as suggested by you, the abbreviation ABI has been written in full and explained before its first use; Secondly, PAD is defined in the last sentence of the paragraph. As follows:

Ankle-brachial index

Systolic blood pressure measurements were taken from the right arm (brachial artery) and both ankles (posterior tibial artery) after a brief period of rest. If the participant's right arm measurements were unavailable due to any conditions that could affect accuracy, such as open wounds or dialysis shunts, the left arm was used for the brachial pressure measurement. Individuals aged 40-59 had their systolic blood pressure measured twice, while those aged 60 and older had it measured only once. The ankle-brachial index (ABI) was calculated by dividing the average systolic blood pressure (ASBP) of the ankle by the ASBP in the arm. For participants aged 60 and older, the single reading represented the mean values. ABI <0.9 was used to define PAD.

Comment 2:

While the methods of the primary study are duly referenced, it is nevertheless necessary to provide a detailed account of the methodology employed for the assessment of vitamin E consumption in this manuscript. Please indicate the number of 24-hour dietary recalls utilized in the study. Please clarify whether vitamin E intake was corrected for energy. It would be beneficial to ascertain whether the participants were taking a vitamin E supplement.

Respond 2:

Thank you for the detailed review. Firstly,based on your this comment, we have added a description of the methodology used to assess vitamin E intake. Furthermore, in this study, the intake of vitamin E was not energy-adjusted, and this limitation has been addressed in the discussion. In addition, data related to vitamin E intake in diets was gathered from the NHANES 1999 to 2004 first-day dietary interview examination files. This interview captured details of all foods and beverages consumed within a 24-hour duration, such as consumption time, eating occasion, food descriptions, portion sizes, food sources, and location of consumption,and did not contain vitamin E supplement . As follows:

Data related to vitamin E intake in diets was gathered from the NHANES 1999 to 2004 first-day dietary interview examination files. A 24-hour dietary recall interview was conducted via the NHANES computer-assisted dietary data interview system to acquire detailed dietary intake data for all participants. This interview captured details of all foods and beverages consumed within a 24-hour duration, such as consumption time, eating occasion, food descriptions, portion sizes, food sources, and location of consumption. Following the dietary recall, a set of health-related inquiries were made. The data collection approach utilized the Automated Multiple Pass Method, which is the dietary data collection tool of the US Department of Agriculture (USDA).

Comment 3:

As noted, Reference 14 examined the same population and investigated the correlation between vitamin E intake and DAP. In what ways does your study differ from that of Reference 14?

Respond 3:

Thank you for the detailed review. The differences between my study and Reference 14 are as follows:

1. My study provides a research flowchart, demonstrating the design rationale of the study;

2. My study includes curve fitting graphs, making the study results more intuitive;

3. The covariates in my study include age, sex, race, marital status, education, physical activity, income, smoking, hypertension, diabetes, cardiovascular disease, body mass index, total cholesterol, and HbA1c, enhancing the reliability of the study;

4. My study further strengthens the stability of the results through subgroup analysis and forest plots. These are all significant improvements that can enhance the scientific rigor and credibility of the study.

Comment 4:

The majority of the references utilized in this study are old, with 80% of them published more than five years ago. It is recommended that more current references be cited, particularly those used in the discussion to support the relationship between vitamin E consumption and DAP.

Respond 4:

Thank you for the valuable suggestion. Based on your advice, the references in the study have been updated and supplemented, with literature published within the last 5 years accounting for 45.2%. However, the literature on the relationship between vitamin E and PAD published in the past 5 years is limited.

2.Zhang Z, Chen Z. Higher Systemic Immune-Inflammation Index is Associated With Higher Likelihood of Peripheral Arterial Disease. Ann Vasc Surg. 2022;84:322–6.

5.Zhu K, Qian F, Lu Q, Li R, Qiu Z, Li L, et al. Modifiable Lifestyle Factors, Genetic Risk, and Incident Peripheral Artery Disease Among Individuals With Type 2 Diabetes: A Prospective Study. Diabetes Care. 2024;47:435–43.

6.Pan D, Guo J, Su Z, Meng W, Wang J, Guo J, et al. Association of prognostic nutritional index with peripheral artery disease in US adults: a cross-sectional study. BMC Cardiovasc Disord. 2024;24:133.

15.Hicks CW, Wang D, Matsushita K, McEvoy JW, Christenson R, Selvin E. Glycated albumin and HbA1c as markers of lower extremity disease inUS adults with and without diabetes. Diabetes

Res Clin Pract. 2022;184:109212.

16.Qu C-J, Teng L-Q, Liu X-N, Zhang Y-B, Fang J, Shen C-Y. Dose-Response Relationship Between Physical Activity and the Incidence of Peripheral Artery Disease in General Population:

Insights From the National Health and Nutrition Examination Survey 1999-2004. Front Cardiovasc Med. 2021;8:730508.

18.Liu Y, Chang L, Wu M, Xu B, Kang L. Triglyceride Glucose Index Was Associated With the Risk of Peripheral Artery Disease. Angiology. 2022;73:655–9.

19.Loreaux F, Jéhannin P, Le Pabic E, Paillard F, Le Faucheur A, Mahe G. An unfavorable dietary pattern is associated with symptomatic peripheral artery disease. Nutr Metab Cardiovasc Dis. 2024;34:2173–81.

20.Liu Y, Wei R, Tan Z, Chen G, Xu T, Liu Z, et al. Association between dietary fiber intake and peripheral artery disease in hypertensive patients. J Health Popul Nutr. 2024;43:118.

24.Cheng Y, Fang Z, Zhang X, Wen Y, Lu J, He S, et al. Association between triglyceride glucose-body mass index and cardiovascular outcomes in patients undergoing percutaneous coronary intervention: a retrospective study. Cardiovasc Diabetol. 2023;22:75.

25.Kumar M, Deshmukh P, Kumar M, Bhatt A, Sinha AH, Chawla P. Vitamin E Supplementation and Cardiovascular Health: A Comprehensive Review. Cureus [Internet]. 2023 [cited 2024 Mar 10]; Available from: https://www.cureus.com/articles/184881-vitamin-e-supplementation-and- cardiovascular-health-a-comprehensive-review.

Comment 5:

In line 228, the text makes reference to "recent studies," yet no corresponding reference is provided. Please provide the references for these recent studies.

Respond 5:

Thank you for the valuable suggestion. This sentence is redundant, it has been deleted.

Comment 6:

The observational, cross-sectional study does not define a causal relationship. It is therefore advisable to exercise caution when considering the phrase "These findings should be used when formulating clinical guidelines."

Respond 6:

Thank you for the valuable suggestion. This sentence is indeed not precise, it has been revised as follows: These findings should be considered when offering dietary guidance to prevent PAD.

Comment 7:

The extant literature indicates that the isolated consumption of vitamin E does not provide cardiovascular protection. This is evidenced by the fact that studies that used supplementation did not observe the same results as observational studies. It is therefore of interest that this topic is introduced for discussion.

Respond 7:

Thank you for the detailed review. I have already supplemented as per your request as follows:

Extensive clinical research has demonstrated the potential protective effects of vitamin E against long-term cardiovascular diseases[28]. Studies have indicated that taking vitamin E supplements for more than four years can result in a substantial 59% decrease in mortality from coronary disease[29]. The discussion regarding the impact of vitamin E on cardiovascular diseases is complex. While certain studies indicate that vitamin E supplements may provide protection against cardiovascular diseases, others do not support this claim. Additionally, a minority of studies have even suggested potential adverse effects[30,31]. So further investigations are necessary to validate our findings and explore the intricate relationships and potential mechanisms involved.

Reviewer #2:

Dear Reviewer,

Thank you very much for your time involved in reviewing the manuscript and your comments have greatly helped to improve the quality of our manuscript.

Comments: It is a well-written but incomplete article. Language editing is strongly recommended. Professional language support is recommended. In addition to the grammar check, the statistical as well as grammatical reasons why certain methods are favoured should be elaborated.

Respond：Thank you for the valuable suggestion. I have made the modifications to the grammatically incorrect parts as per your suggestion. Thank you once again for your guidance. After referring to a large number of previous literatures, the statistical methods used in this study include curve fitting, single factor analysis, multiple factor analysis, stratified analysis, subgroup analysis, etc.

1. Through the interpolation algorithm, curve fitting can generate a smooth curve between the given data points, thereby better understanding the trend and pattern of the data；

2. Single factor analysis not only helps in the initial exploration of the relationship between the predictor and response variables, aiding researchers in understanding the impact of individual variables on the outcome. It also assists researchers in selecting the variables that need to be included in the multiple factor analysis, for instance, by evaluating the changes in partial regression coefficients or OR values.

3. Univariate analysis primarily explores the impact of a single independent variable on the dependent variable through t-tests or analysis of variance (ANOVA). The t-test is suitable for comparing the means of two independent samples or paired samples, while ANOVA is used for comparing the means of three or more groups. These two methods are concise and straightforward, and can intuitively reveal the significant impact of a specific factor when acting independently on the outcome.

4. Multiple factor analysis uses multiple regression analysis to simultaneously examine the combined impact of multiple independent variables on the dependent variable and the interaction effects among the independent variables. This method can more comprehensively reflect the actual situation and reveal the hidden relationships behind complex data. Through multiple regression analysis, we can obtain the regression coefficients of each independent variable, thereby quantifying its specific contribution to the dependent variable, providing strong support for in-depth research.

5. Stratified analysis involves dividing the study subjects into strata based on one or more variables that need to be controlled, which can reduce the impact of confounding factors on the research results and thus provide a more accurate estimation of the association between the exposure factor and the outcome variable.

6. The main purpose of subgroup analysis is to study interactions or effect modification, i.e., whether the effect size varies in different populations or under different conditions. By dividing the study subjects into different subgroups based on certain characteristics (such as gender, disease severity, etc.), estimating the effect sizes for each subgroup separately, and conducting comparisons between subgroups, a better understanding of specific patient, intervention type, or study-specific issues can be achieved.

---

## [Decision Letter · Decision Letter 1]

17 Jan 2025

PONE-D-24-23667R1Association of dietary vitamin E intake with peripheral arterial disease: A retrospective cross-sectional studyPLOS ONE

Dear Dr. Liu,

Thank you for submitting your manuscript to PLOS ONE. After careful consideration, we feel that it has merit but does not fully meet PLOS ONE’s publication criteria as it currently stands. Therefore, we invite you to submit a revised version of the manuscript that addresses the points raised during the review process.

We look forward to receiving your revised manuscript.

Kind regards,

Hasan Durmus

Academic Editor

PLOS ONE

Journal Requirements:

Reviewers' comments:

Reviewer's Responses to Questions

**Comments to the Author**

1. If the authors have adequately addressed your comments raised in a previous round of review and you feel that this manuscript is now acceptable for publication, you may indicate that here to bypass the “Comments to the Author” section, enter your conflict of interest statement in the “Confidential to Editor” section, and submit your "Accept" recommendation.

Reviewer #1: All comments have been addressed

Reviewer #2: All comments have been addressed

2. Is the manuscript technically sound, and do the data support the conclusions?

Reviewer #1: Yes

Reviewer #2: Yes

3. Has the statistical analysis been performed appropriately and rigorously? 

Reviewer #1: Yes

Reviewer #2: Yes

4. Have the authors made all data underlying the findings in their manuscript fully available?

Reviewer #1: Yes

Reviewer #2: Yes

5. Is the manuscript presented in an intelligible fashion and written in standard English?

Reviewer #1: Yes

Reviewer #2: Yes

6. Review Comments to the Author

Reviewer #1: (No Response)

Reviewer #2: Dear Author

I've checked your corrections. I still have two suggestions. Firstly, please review the journal's reference citation rules. (Reference 4 is not appropriate.) Secondly, in the conclusion section, not only nutrition guidelines can be emphasised but also nutrition education can be emphasised.

7. PLOS authors have the option to publish the peer review history of their article (what does this mean? ). If published, this will include your full peer review and any attached files.

**Do you want your identity to be public for this peer review?** For information about this choice, including consent withdrawal, please see our Privacy Policy .

Reviewer #1: No

Reviewer #2: No

---

## [Author Response · Author response to Decision Letter 1]

4 Feb 2025

Responds to the reviewer’s comments:

Reviewer #1: (No Response)

Reviewer #2:

Dear Author

I've checked your corrections. I still have two suggestions. Firstly, please review the journal's reference citation rules. (Reference 4 is not appropriate.) Secondly, in the conclusion section, not only nutrition guidelines can be emphasised but also nutrition education can be emphasised.

Respond：Thank you for the detailed review. Firstly,based on your this comment, we have revised the format of the fourth reference according to the citation rules of the journal.

Secondly, the conclusion section has also been revised according to your suggestions.

1.Martin SS, Aday AW, Almarzooq ZI, Anderson CAM, Arora P, Avery CL, et al. 2024 Heart Disease and Stroke Statistics: A Report of US and Global Data From the American Heart Association. Circulation. 2024;149:e347–913.

2.These findings should be considered when offering dietary guidance and nutrition education to prevent PAD.

---

## [Decision Letter · Decision Letter 2]

18 Feb 2025

Association of dietary vitamin E intake with peripheral arterial disease: A retrospective cross-sectional study

PONE-D-24-23667R2

Dear Dr. Liu,

We’re pleased to inform you that your manuscript has been judged scientifically suitable for publication and will be formally accepted for publication once it meets all outstanding technical requirements.

Kind regards,

Hasan Durmus

Academic Editor

PLOS ONE

Additional Editor Comments (optional):

I would like to inform you that your revised manuscript has undergone a thorough evaluation, and I find the changes to be satisfactory in addressing the key concerns raised during the peer review process. At this stage, the manuscript has been recommended for further consideration by the journal’s editorial team. The final decision will be made following additional editorial checks and approval from the Editor-in-Chief.

We appreciate your patience and efforts in improving the quality of the manuscript. Please feel free to reach out if you have any questions regarding the process.

Reviewers' comments:

Reviewer's Responses to Questions

**Comments to the Author**

1. If the authors have adequately addressed your comments raised in a previous round of review and you feel that this manuscript is now acceptable for publication, you may indicate that here to bypass the “Comments to the Author” section, enter your conflict of interest statement in the “Confidential to Editor” section, and submit your "Accept" recommendation.

Reviewer #2: All comments have been addressed

2. Is the manuscript technically sound, and do the data support the conclusions?

Reviewer #2: Yes

3. Has the statistical analysis been performed appropriately and rigorously? 

Reviewer #2: Yes

4. Have the authors made all data underlying the findings in their manuscript fully available?

Reviewer #2: Yes

5. Is the manuscript presented in an intelligible fashion and written in standard English?

Reviewer #2: Yes

6. Review Comments to the Author

Reviewer #2: Dear Author,

First of all, congratulations on the acceptance of your work. The topic of your work is very topical and makes an important contribution to the literature.

7. PLOS authors have the option to publish the peer review history of their article (what does this mean? ). If published, this will include your full peer review and any attached files.

**Do you want your identity to be public for this peer review?** For information about this choice, including consent withdrawal, please see our Privacy Policy .

Reviewer #2: No

---

## [Editor Report · Acceptance letter]

PONE-D-24-23667R2

PLOS ONE

Dear Dr. Liu,

I'm pleased to inform you that your manuscript has been deemed suitable for publication in PLOS ONE. Congratulations! Your manuscript is now being handed over to our production team.

Kind regards,

on behalf of

Dr. Hasan Durmus

Academic Editor

PLOS ONE